# The Integration of Micro-CT Imaging and Finite Element Simulations for Modelling Tooth-Inlay Systems for Mechanical Stress Analysis: A Preliminary Study

**DOI:** 10.3390/jfb16070267

**Published:** 2025-07-21

**Authors:** Nikoleta Nikolova, Miryana Raykovska, Nikolay Petkov, Martin Tsvetkov, Ivan Georgiev, Eugeni Koytchev, Roumen Iankov, Mariana Dimova-Gabrovska, Angela Gusiyska

**Affiliations:** 1Department of Conservative Dentistry, Faculty of Dental Medicine, Medical University Sofia, 1431 Sofia, Bulgaria; a.gusijska@fdm.mu-sofia.bg; 2Institute of Information and Communication Technologies, Bulgarian Academy of Sciences, 1113 Sofia, Bulgaria; miryana.raykovska@3dlab.iict.bas.bg (M.R.); nikolay.petkov@3dlab.iict.bas.bg (N.P.); ivan.georgiev@parallel.bas.bg (I.G.); 3Centre of Excellence in Informatics and Information and Communication Technologies, 1113 Sofia, Bulgaria; 4Institute of Mathematics and Informatics, Bulgarian Academy of Sciences, 1113 Sofia, Bulgaria; m.tsvetkov@math.bas.bg; 5Institute of Mechanics, Bulgarian Academy of Sciences, 1113 Sofia, Bulgaria; koytchev@imbm.bas.bg; 6National Centre of Mechatronics and Clean Technologies, 1113 Sofia, Bulgaria; 7Department of Prosthetic Dentistry, Faculty of Dental Medicine, Medical University Sofia, 1431 Sofia, Bulgaria; m.dimova@fdm.mu-sofia.bg

**Keywords:** computational modelling, dental inlay, dental onlay, finite element analysis, micro-CT

## Abstract

This study presents a methodology for developing and validating digital models of tooth-inlay systems, aiming to trace the complete workflow from clinical procedures to simulation by involving dental professionals—dentists for manual cavity preparation and dental technicians for restoration modelling—while integrating micro-computed tomography (micro-CT) imaging with finite element analysis (FEA). The proposed workflow includes (1) the acquisition of high-resolution 3D micro-CT scans of a non-restored tooth, (2) image segmentation and reconstruction to create anatomically accurate digital twins and mesh generation, (3) the selection of proper resin and the 3D printing of four typodonts, (4) the manual preparation of cavities on the typodonts, (5) the acquisition of high-resolution 3D micro-CT scans of the typodonts, (6) mesh generation, digital inlay and onlay modelling and material property assignment, and (7) nonlinear FEA simulations under representative masticatory loading. The approach enables the visualisation of stress and deformation patterns, with preliminary results indicating stress concentrations at the tooth-restoration interface integrating different cavity alternatives and restorations on the same tooth. Quantitative outputs include von Mises stress, strain energy density, and displacement distribution. This study demonstrates the feasibility of using image-based, tooth-specific digital twins for biomechanical modelling in dentistry. The developed framework lays the groundwork for future investigations into the optimisation of restoration design and material selection in clinical applications.

## 1. Introduction

Among the various therapeutic options, indirect restorations (inlays, onlays, overlays) have gained prominence for their ability to restore both function and aesthetics while preserving tooth structure [1]. Specifically, inlays are a durable and conservative alternative to full crowns, allowing for greater preservation of natural tooth tissue [2,3]. Despite these advantages, accurately evaluating the long-term mechanical performance and durability of dental inlay systems presents significant challenges. While controlled clinical trials remain the gold standard for assessing dental materials [4], recent systematic reviews on FEA in dental applications consistently highlight the need for more precise, patient-specific biomechanical insights to complement in vivo studies. FEA has become a pivotal tool in dentistry and biomedical sciences, enabling the simulation of complex biomechanical behaviour in anatomically accurate, patient-specific models. In dentistry, FEA is successfully used in various areas, including

–The analysis of stress and strain distribution;–The evaluation of implant design and performance;–The assessment of new dental materials;–The optimisation of prosthetic restorations;–The study of orthodontic tooth movement [5,6].

Recent studies demonstrate that the integration of advanced imaging technologies—particularly micro-computed tomography (micro-CT) and cone-beam computed tomography (CBCT)—has significantly enhanced the realism and clinical relevance of FEA models. These technologies provide high-fidelity anatomical data for comprehensive biomechanical assessment. The literature also underscores the limitations of conventional modelling approaches in accurately representing complex anatomical structures and their interactions with restorative materials under realistic loading conditions, further highlighting the need for advanced computational methodologies. Consequently, advanced imaging and simulation techniques are urgently needed to provide a deeper understanding of the mechanical behaviour of a tooth with an adhesively luted indirect restoration, hereafter referred to as “tooth-inlay systems.” Micro-CT imaging provides an unparalleled high-resolution representation of dental anatomy, allowing for a detailed visualisation of structures at a microscopic level [7]. Integrating micro-CT with finite element simulations marks a significant advancement in dental science [8], particularly for analysing and optimising tooth-inlay systems. This optimisation often prioritises minimising stress levels in the tooth under maximum loads, a critical condition for reducing the risk of structural failure in treated teeth. This transformative approach enhances the efficiency and effectiveness of restorative dental practices. By converting precise micro-CT scans into geometrical objects, researchers can conduct finite element simulations that accurately predict system behaviour under various loading conditions. The ability to transform a real object into a highly accurate digital geometric model is an essential prerequisite for defining an adequate mathematical model, enabling dental professionals to explore stress distribution and mechanical behaviour. This predictive modelling offers invaluable insights into potential failure points and structural integrity, areas that were previously challenging to assess [9,10].

A standout benefit of this integrated methodology is its remarkable time efficiency. Unlike traditional labour-intensive physical testing and empirical assessments of dental restorations, combining micro-CT imaging and FEA enables rapid virtual evaluations. This efficiency allows researchers to simulate multiple scenarios quickly, evaluating diverse inlay geometries and material properties without the need for extensive physical prototypes. Such an approach significantly accelerates research analysis, reduces resource costs, and makes conducting more comprehensive studies feasible within shorter timeframes. This integration streamlines the analysis process for various restorative materials and cavity forms, ensuring research practicality and cost-effectiveness. Furthermore, this approach enables a design optimisation strategy which involves identifying candidate materials with specific mechanical characteristics to achieve predefined performance requirements for the tooth-inlay system. The high-fidelity digital models generated through this methodology also enable virtual assessments of potential risks associated with tooth wall failures under diverse loading conditions [11]. By simulating stress concentrations within the tooth structure, researchers can identify critical areas susceptible to long-term damage [12]. This capability not only aids in understanding planned or existing restorations but also inspires future design improvements, ultimately enhancing the longevity and effectiveness of restorative procedures.

The aim of this study is to develop and validate an integrated digital workflow for generating anatomically accurate tooth-specific models of tooth-inlay systems by combining micro-CT imaging, 3D printing, manual cavity preparation performed by a dentist, digital restoration modelling by a dental technician, and FEA to enable the comparative mechanical evaluation of different restoration designs on the same tooth geometry.

This study addresses the following question: how can a clinically informed digital workflow—integrating micro-CT imaging, physical prototyping, dentist-led cavity preparation, technician-designed restorations, and FEA—be developed to generate validated, tooth-specific digital twins for evaluating the mechanical behaviour of indirect restorations?

We hypothesise that the proposed workflow will produce high-fidelity digital models capable of revealing clinically relevant mechanical behaviours, such as stress concentrations at restoration interfaces, under simulated masticatory loading.

## 2. Materials and Methods

### 2.1. Selection of Tooth

As part of this study, we used a human second molar tooth with a fused root as a specimen that had been extracted due to surgical issues and subsequently subjected to mechanical cleaning using a periodontal curette to remove organic debris, followed by polishing with a brush and paste. The specimen was stored at 4 °C in a 0.1% thymol solution prepared in physiological saline (pH 7) to maintain its integrity. A stereomicroscopic examination at 40× magnification (Leika S6, Leika Microsystem GmbH, Wetzlar, Germany) was conducted to identify and exclude specimens exhibiting fractures or cracks in the coronal and radicular portion. Written informed consent was obtained from the patient to use the tooth for this in vitro investigation and to publish the results in this paper.

### 2.2. Preparation/Fabrication/Creation of Typodont Models and Obturating Constructions

#### 2.2.1. First Micro-CT Scanning

The specimen was scanned using a Nikon XT H 225 (Nikon Metrology, Tring, UK) system, which captured 2525 digital radiographic projections at a voltage of 100 kV and a tube current of 110 µA, 700 ms exposure time, and 1 mm Al filtration of the beam. The resulting resolution had a cubic dimension of 10 × 10 × 10 μm. The scanning process utilized Inspect-X, version XT 3.1.3 (Nikon Metrology, Tring, UK) for acquisition. The reconstruction software was X-AID v2023.11.1 (MITOS GmbH, Garching, Germany). Segmentation, surface model refinement, and extraction were performed using VGSTUDIO MAX 2023.4 (Volume Graphics GmbH, Heidelberg, Germany). The model was optimised using Meshmixer (Autodesk, San Francisco, CA, USA). However, part of the natural anatomy significantly increased the surface model’s complexity and the need for computing resources. The calcified regions were digitally three-dimensionally segmented and removed, as they were considered not essential for the intended mechanical analysis. Masking of these regions was conducted with a region-growing algorithm and a digital pen, and regions were filled with grey values matching adjacent pulp space to maintain anatomical continuity.

For the surface irregularities and morphological protrusions in the root canal that would have unnecessarily complicated the final model, a filtering algorithm was applied. The filter of choice was Taubin smoothing.

#### 2.2.2. Printing the Typodonts

As part of the digital twin creation workflow, typodonts were fabricated using the Anycubic Photon Mono 2 3D printer (ANYCUBIC TECHNOLOGY, Hong Kong, China), which employs Masked Stereolithography Apparatus (MSLA) technology—a variant of LCD-based photo-polymerisation. This method integrates a UV light source with a monochrome LCD screen acting as a dynamic mask, selectively curing photopolymer resin layer by layer.

Printing was conducted at a layer height of 50 μm using Anycubic Water-Wash Resin + Grey (ANYCUBIC TECHNOLOGY, Hong Kong, China), selected for its favourable mechanical properties and aesthetic appearance—critical for the subsequent manual cavity preparations. Models were strategically arranged on the build platform to maximise spatial efficiency, and appropriately designed supports were added to ensure structural stability and dimensional accuracy throughout the fabrication process.

#### 2.2.3. Preparation of Cavities

To explore the influence of cavity geometry on the same tooth and different types of cavities, three samples were prepared conventionally, while one was prepared to employ biomimetic preparation onto the four typodont models. This preparation was performed using a high-speed handpiece (NSK PANA-MAX, Kanuma, Japan). To ensure meticulous precision and reproducibility of the varied cavity geometries, this procedure was conducted under magnification provided by a dental operating microscope (Dental microscope Opmi Pico 6x, Carl Zeiss, Oberkochen, Germany).

#### 2.2.4. Second Micro-CT Scanning (Post-Cavity Preparation)

Following cavity preparation, the four typodont models were subjected to a micro-CT scan using the Nikon XT H 225 system (Nikon Metrology, UK). This re-scanning phase is conducted with different settings, which were determined by the material of the typodonts, and it captured 2525 digital radiographic projections at a voltage of 80 kV and a tube current of 100 µA with an exposure time of 700 ms per projection. Similarly to the initial scan, the reconstructed images yielded an isotropic voxel size of 10 × 10 × 10 µm. This second scan was critical for obtaining the precise post-preparation geometry for each unique cavity.

#### 2.2.5. Virtual Design of Indirect Restorations

The STL files representing the prepared typodonts were imported into Exocad dental software (Exocad DentalCAD (Version 2.4), Exocad GmbH, https://exocad.com (accessed on 10 March 2025)) for the virtual design of indirect inlay restorations. A 100 µm cement space was defined during this process, and occlusal surfaces were standardised based on the original prototype molar.

### 2.3. Preparation of Models for FEA

In 3ds Max (Autodesk 3ds Max (Version 2023). Autodesk Inc. https://www.autodesk.com/products/3ds-max (accessed on 10 March 2025)), the following three key actions were performed to finalise the digital models:A cement layer was created using the “Boolean” function based on the predefined gap between the tooth structure and the obturating construction.A base element simulating the supporting bone was modelled to provide appropriate boundary conditions during simulation.

Since the 3D printed typodonts lacked internal anatomical features such as the root canal system, the segmented canal structure obtained from the micro-CT scan of the natural tooth was digitally merged with the typodont geometry to produce an anatomically enhanced model for simulation. All digital components—including four tooth models with integrated endodontic systems and prepared cavities, four obturating structures, four cement layers, and four supporting bone models—were exported from 3ds Max in STL format, preserving their original coordinates. These files were sequentially imported into FreeCAD (FreeCAD Version 0.20.2, 2023, https://www.freecadweb.org, accessed on 25 March 2025), where they were converted into solid volumetric models in STEP format. The conversion process involved generating shapes from meshes with a tolerance of 0.01 mm, refining the geometry, and exporting the solids for further simulation use. The complete assemblies were then imported into ANSYS (ANSYS Version 2022R2, ANSYS, Inc., Canonsburg, PA, USA) for finite element analysis (FEA). This stage involved simulating the biomechanical response of the tooth-inlay systems under representative loading conditions.

### 2.4. Finite Element Analysis

The FEA was conducted to investigate the stress distribution within tooth-inlay systems reconstructed from micro-CT data. The main elements of the simulation workflow are described below.

Geometrical Model Discretisation: The 3D geometry of the tooth inlay from the STEP file was imported into the ANSYS pre-processor and discretised using tetrahedral second-order finite elements. Mesh refinement was applied in critical regions, particularly near interfaces between materials, to capture potential stress gradients and ensure numerical stability.Material Properties and Constitutive Laws: Each domain in the model (enamel, dentin, inlay material, and adhesive interface) was assigned material properties obtained from published experimental data. A linear elastic, isotropic constitutive model was used for all materials under the assumption of small strains and negligible time-dependent effects. Material parameters such as Young’s modulus and Poisson’s ratio were defined individually for each component.Boundary and Initial Conditions: To replicate physiological conditions, the base of the tooth was constrained in all directions to simulate its anchorage within the alveolar socket. A vertical compressive load was applied to a selected location on the occlusal surface to mimic masticatory loading. The load magnitude was selected based on the upper range of functional bite forces observed in fully dentate individuals. The load application point was chosen based on anatomical relevance and the location of expected peak stress. A static linear analysis was performed, assuming quasi-static loading conditions and neglecting dynamic effects [13,14,15].Contact Modelling: The interfaces between dissimilar materials—such as between the dentin and inlay or enamel and adhesive—were modelled assuming perfect (full) contact, i.e., without interfacial separation or sliding. This assumption implies ideal bonding and stress continuity across interfaces, allowing for a simplified representation of load transfer between materials. While this approach may not capture all interfacial failure mechanisms, it provides an initial approximation for evaluating internal stress distributions. This modelling strategy enabled a detailed assessment of stress distributions within the tooth–restoration assembly and the identification of zones with elevated mechanical risk under functional loading.

## 3. Results

The high-resolution industrial micro-CT scanning of the extracted human molar provided detailed digital datasets with reduced quantum noise and beam-hardening artefacts. The analysis of the computed tomography data revealed multiple calcified structures within the pulp chamber and root canals (Figure 1). While anatomically accurate, these features increased model complexity and computational demands. Consequently, calcified regions were digitally segmented and removed, as they were not essential for the intended mechanical analysis. This masking was achieved using a region-growing algorithm and manual refinement with a digital pen, followed by a grey value interpolation to preserve anatomical continuity.

A further inspection of the segmented root canal system revealed morphological irregularities and surface noise that would have compromised the final mesh. To address this, a Taubin smoothing algorithm was applied to the preliminary mesh [16,17]. This filter effectively reduced geometric complexity while preserving surface integrity and preventing mesh shrinkage (Figure 2). A clean surface mesh was generated and exported in STL format. Initial micro-CT scanning of the tooth specimen yielded a high-resolution polygonal model comprising 504 and 300 triangles. The application of Meshmixer effectively addressed various mesh defects, with many minor imperfections automatically corrected. More complex topological errors, however, necessitated targeted manual intervention. Following these corrective procedures, the polygon count was systematically reduced to approximately 30,000 triangles. This decimation process was meticulously monitored to ensure the precise preservation of all critical anatomical features, maintaining their clear expression for accurate downstream analysis.

The typodont models were successfully 3D printed, and cavity preparations were performed by an experienced dentist. The resin was selected for its favourable mechanical properties and high glass transition temperature, ensuring that the model remained stable and did not deform during preparation with a high-speed handpiece and dental bur. Three models received conventional preparations, while one followed a biomimetic approach. After preparation, the printed models were rescanned and updated meshes were generated. The STL files of the four prepared typodonts were imported into Exocad, where a dental technician designed the corresponding indirect restorations. During this step, a cement gap of 100 microns was defined, and occlusal surfaces were standardised using the original natural occlusal surface of the scanned prototype. To complete the assembly, the cement layer was modelled in 3ds Max using the Boolean function based on the remaining space between the tooth and the restoration. A comprehensive visualisation of the assembled digital models is presented in Figure 3. This figure comprehensively displays the four distinct types of cavity preparations investigated in this study alongside their respective obturation structures, as well as the precisely modelled cement layers interconnecting the tooth and restoration components for each variation.

Additionally, a supporting base element has been designed to replicate the surrounding bone tissue and define mechanical boundary conditions, as depicted in Figure 4b. For the finite element analysis, the prepared digital models underwent a transformation into an FEA-ready format. This was achieved by exporting the models in STL format, importing them into FreeCAD, and converting them into solid volumetric STEP files using a mesh tolerance of 0.01 mm. The result is comprehensively illustrated in Figure 4. Figure 4a details the final geometry of the solid bodies, precisely representing the obturation, the cement layer, and the meticulously rendered root canal system. Building upon this geometric foundation, Figure 4b presents the corresponding finite element model of the obtained solid bodies with the tooth-supporting bone [18].

The presentation of the finite element model of the tooth-inlay system, along with the resulting displacement field and von Mises stress distribution under the applied loading conditions—a 600 N axial compressive load on the occlusion surface in a selected small area (set of surface elements) around point B—is illustrated in Figure 5. Figure 5b,c shows that the developed workflow can generate mechanically consistent outputs suitable for assessing the structural response of the restoration. The results are shown for the following specific material parameters: dentin and enamel materials are modelled with a Young’s modulus of 18.6 GPa and Poisson ratio of 0.31; the inlay material has a Young’s modulus of 15 GPa (e.g., resin composite) and Poisson ratio of 0.28; the luting cement has a Young’s modulus of 12 GPa and Poisson ratio of 0.25; and the cortical bone has a Young’s modulus of 13.7 GPa and Poisson ratio of 0.3 [19,20,21]. The displacement and stress maps highlight the deformation patterns and stress concentrations within the restored structure, offering insights into the mechanical behaviour of the modelled configuration.

## 4. Discussion

Creating a precise model suitable for numerical simulation is a multistep process requiring expertise in various tools and software platforms. In this study, each stage of the workflow was carefully optimised to ensure anatomical accuracy and computational efficiency. Clear image acquisition during industrial micro-CT scanning was achieved by following established protocols, such as beam filtration with metal filters, median filtering during reconstruction, longer exposure times, and increased image counts. These measures effectively reduced quantum noise and beam-hardening artefacts, improving the signal-to-noise ratio and image fidelity [22]. The segmentation of dental structures, particularly the root canal system, posed notable challenges due to morphological irregularities and anatomical complexity [23]. In our workflow, the use of a region-growing algorithm for dentin segmentation followed by region-of-interest (*ROI*) refinement proved effective in isolating essential structures and simplifying complex geometries. This approach enhanced both the accuracy and manageability of the final mesh. The efficacy of such segmentation strategies has been corroborated by recent studies [24,25]. Several authors, including Rodrigues et al., Magne, Wicaksono et al., and Tekin et al., have used micro-CT imaging to generate highly detailed dental geometries [26,27,28,29]. These studies emphasised the importance of mesh optimisation, particularly polygon count reduction, to balance anatomical fidelity with computational feasibility. For similar reasons, we used Meshmixer—chosen for its user-friendliness and free accessibility—to repair mesh errors and reduce triangle density. Model simplification was carefully considered. Chen et al. designed Class II cavity geometries in micro-CT-derived models to study deep margin elevation while preserving anatomical integrity [30]. In our study, cavity preparations were performed manually by a dentist, ensuring clinically relevant morphology and replicating real-world operative techniques. The modelling of the cement layer in our study was inspired by Wicaksono et al., who explicitly modelled adhesive layers to simulate stress transfer at the tooth–restoration interface [29]. In contrast, our cement layer was digitally extracted using a Boolean function to reflect the space defined by the dental technician during restoration design. Tekin et al. advanced micro-CT modelling by including full assemblies of crown, post, periodontal ligament (PDL), and alveolar bone [28]. In our protocol, a dental technician digitally modelled the restorations using dedicated CAD software. Given the current digitisation of dental laboratory workflows, incorporating technician-designed restorations reflects routine clinical practice and ensures realism in the modelled structures. Due to the limitations in 3D printer resolution, it was not feasible to replicate detailed root canal anatomy in the printed typodonts. To overcome this, we transferred the canal system from the initial micro-CT scan to the updated post-preparation models. This decision preserved critical anatomical details, which are known to influence the biomechanical response of the tooth under load.

The limitations of the study are as follows:Adjacent structures are omitted.The current model excludes the periodontal ligament (*PDL*), neighbouring teeth, and soft tissues. These structures may significantly influence stress distribution and displacement behaviour under clinical loading.Polygon reduction effects are present.Although polygon count reduction was necessary to improve computational efficiency, it may have resulted in the loss of fine anatomical details in localised regions.Micro-CT applicability was limited.While micro-CT provides exceptional image resolution, it is limited to ex vivo use. For clinical translation, the workflow must be adapted for lower-resolution CBCT, which may reduce model accuracy.

### Clinical Applicability and Future Directions

Despite the limitations, the proposed workflow can be adapted for use with CBCT data. Zheng et al. reviewed segmentation methods for CBCT and highlighted the value of region-growing techniques for delineating complex dental anatomies. These methods benefit from anatomical priors and can accommodate morphological variability, supporting their integration into clinical modelling pipelines [31]. Other researchers, such as Camargos et al. and Özcan et al., favoured CBCT-derived models for their clinical accessibility [32,33]. Camargos et al. demonstrated that CBCT, combined with intelligent mesh optimisation and CAD-based reconstruction, can yield FEA-compatible models [32]. Özcan et al. further integrated patient-specific mandibular kinematics using Modjaw^®^ motion capture, introducing dynamic boundary conditions that simulate realistic occlusal function [33]. Hasegawa et al. created whole-jaw models using CT data and achieved detailed mesh tuning across tissues like enamel and cortical bone [34]. To support wider clinical adoption, some studies, such as those by Camargos and Lahoud et al., proposed low-cost modelling strategies using open source software and automation [32,35]. Lahoud et al. emphasised that CBCT-based machine learning-assisted modelling is critical to advancing personalised dentistry [35]. When combined with our workflow, these techniques offer a clear path toward fully patient-specific modelling, bridging the gap between high-resolution research and routine clinical practice.

## 5. Conclusions

This study presents a comprehensive workflow for generating finite element models of tooth-inlay systems, starting from natural tooth structures and progressing to digitally reconstructed and 3D printed typodonts with varying cavity designs and restorative strategies. The workflow integrates high-resolution micro-CT imaging, computer-aided design, additive manufacturing, and expert input from dental professionals to produce anatomically accurate and clinically relevant digital models. To demonstrate the feasibility and robustness of the proposed workflow, one representative inlay configuration was selected for finite element analysis. The simulation results confirm the applicability of the digital workflow for biomechanical assessment. Although multiple inlay configurations were prepared, this paper focuses on a single case as a proof of concept. A comparative analysis of various inlay geometries and materials is planned for future studies to support evidence-based restorative design.

Future work will focus on applying the proposed workflow to optimise restorative strategies by incorporating both extracted teeth and clinical scans acquired during actual restorative procedures, enabling patient-specific modelling and simulation.

## Figures and Tables

**Figure 1 jfb-16-00267-f001:**
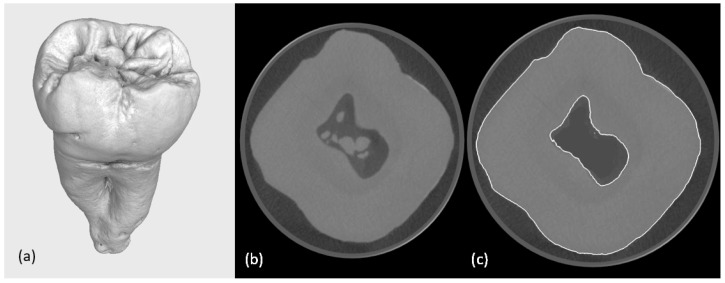
(**a**) A reconstruction of a second human molar; (**b**) slice from the micro-CT scanning calcifications are visible in the pulp chamber; (**c**) tooth with digital surface determination and digital removal of calcifications from the pulp chamber.

**Figure 2 jfb-16-00267-f002:**
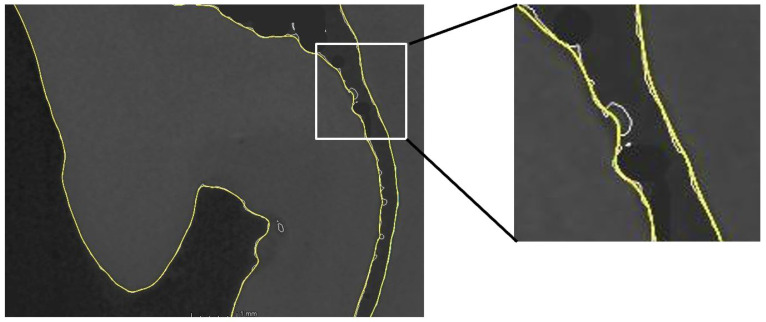
Demonstration of the refinement. Situation before (white contour surface determination) and situation after (yellow contour).

**Figure 3 jfb-16-00267-f003:**
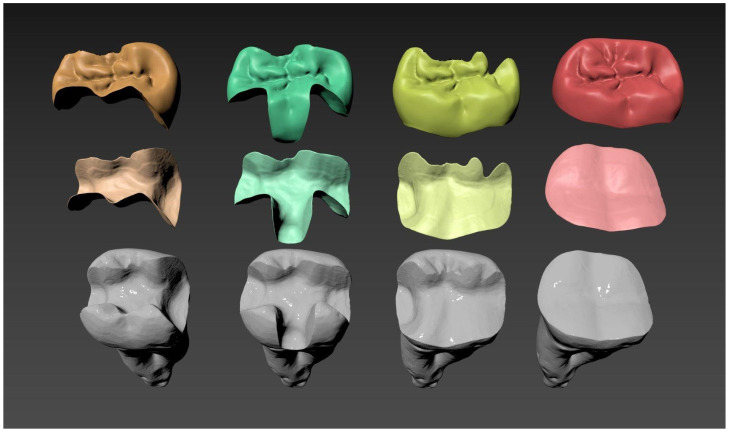
Visualisation of the four types of cavities, the obturation structures, and the cement layers for each.

**Figure 4 jfb-16-00267-f004:**
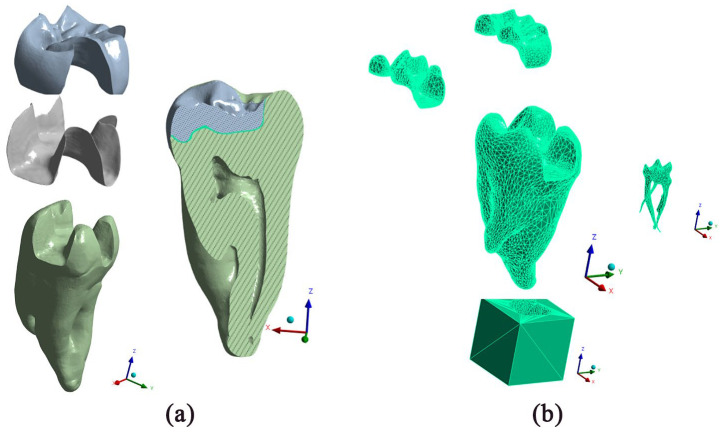
Finite element model generation. (**a**) Final solid geometry of the assembled digital model, depicting the obturation, cement layer, the inlay, and tooth structure (canal) and (**b**) the resulting finite element mesh with the supporting base element prepared for numerical simulation in ANSYS.

**Figure 5 jfb-16-00267-f005:**
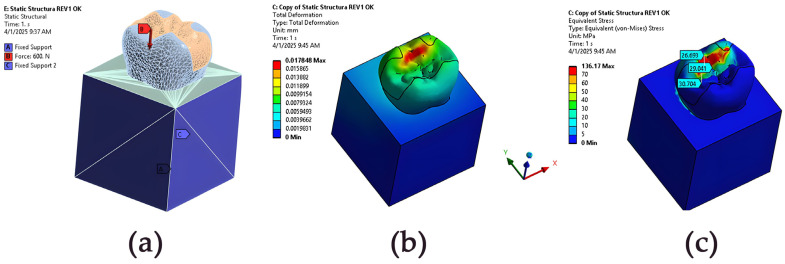
ANSYS finite element analysis results for a tooth-inlay system: (**a**) representative FE model of the restored tooth, illustrating the applied boundary conditions and masticatory loading; (**b**) displacement magnitude map showing the deformation patterns (total displacements in mm) under masticatory loading; (**c**) contour map of von Mises stress distribution, highlighting critical stress concentrations at the tooth–restoration interface. Units are [MPa] for stress and [mm] for displacement.

## Data Availability

The data presented in this study are available upon request from the corresponding author.

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
