# Peer review of "The Integration of Micro-CT Imaging and Finite Element Simulations for Modelling Tooth-Inlay Systems for Mechanical Stress Analysis: A Preliminary Study"

_jfb, 2025, doi:10.3390/jfb16070267_

Round 1
Reviewer 1 Report
Comments and Suggestions for Authors
Please find the uploaded file.

minor grammar issues
Author Response
|
Comments 1: The title needs to be simplified and remove the repletion like repeating the word for. I suggest this title: Integrating Micro-CT Imaging and Finite Element Simulations to Model Tooth-Inlay Systems for Mechanical Stress Analysis: A Preliminary Study |
|
Response 1: Thank you for your valuable suggestion regarding the manuscript title. We appreciate your recommendation to simplify the wording and remove the repetition. In response, we have revised the title accordingly. The updated title now reads: "Integrating Micro-CT Imaging and Finite Element Simulations to Model Tooth-Inlay Systems for Mechanical Stress Analysis: A Preliminary Study" We believe this version improves clarity and conciseness, and we are grateful for your input in helping us refine the presentation of our work. This change can be found – page 1, line 2.
|
|
Comments 2: About Abstract: 1.Please add the methodology steps and add data and conclusion. Just stating only, the objective and combining two techniques to assess the mechanical performance of dental indirect restorations is not very informative! 3. Please revise your keywords to be aligned with MeSH database for better visibility
|
|
Response 2: Thank you for your constructive feedback on the abstract. In response:
This change can be found – page 1, from line 21.
|
|
Comments 3: About Introduction: Generally, it is well structured, and research problem clear. However,
Response: 3: Thank you for your insightful comments regarding the Introduction. In response: 1. We have added references to recent and relevant literature on FEA in dental applications that directly support the aim of our study. While a comprehensive review of FEA was not the focus, this contextual information strengthens the rationale behind the workflow we propose. 2. The research question has been clearly stated to define the scope of the study. 3. The aim has been revised to emphasize the development and validation of a digital modeling workflow. Additionally, a hypothesis has been included to outline the expected functionality of the proposed method. This change can be found – page 2, from line 43
Comments 4: About Methodology 1-Only four cavity types were simulated, and only one natural tooth was used as the anatomical prototype. This raises concerns about generalizability. In addition, justification for using a single prototype should be clearly stated. Response: 4: Thank you for your constructive comments on the methodology. In response: 1. We have revised the Introduction to clarify that the primary aim of this study is to present and validate a specific workflow for obtaining informative and reliable 3d model for numerical simulation in dentistry. For this reason, a single natural tooth was intentionally selected as a consistent anatomical prototype to demonstrate the feasibility of the proposed approach. 2. The Materials and Methods section has been substantially revised to include all relevant details—such as scanning parameters, segmentation techniques, printing settings, and material properties—to ensure that the workflow can be replicated by other researchers. 3. As the study is focused on demonstrating the modeling workflow rather than performing comparative or inferential analysis, statistical testing was not applicable at this stage. This change can be found – page 3, from line 120
Comments 5: About Result - Human molars extracted for periodontal reasons: Scanning completed for one molar or more than this. - Figures must be fully interpreted in detail within the context. text. for the four inlay types would greatly enhance clarity.
Response: 5: Thank you for your detailed observations regarding the Results section. In response: · The use of the plural “molars” was a typographical error and has been corrected to reflect that only one natural second molar was used in this study. · Figure captions have been extended to provide clearer interpretation and integration with the text, as recommended. · Figure 5 has been updated to improve clarity and better align its caption with the content and discussion in the main text. · As the focus of this study is on presenting and validating the synergetic workflow, the finite element analysis was discussed qualitatively. At this preliminary stage, numerical comparisons between inlay types were not the aim. A detailed, quantitative FEA comparison will be part of future work. This change can be found – page 6, from line 251
Comments 6: About Discussion and conclusion - what is the limitation of this study? Response: 6: Thank you for your thoughtful comments on the Discussion and Conclusion sections. In response: We acknowledge the suggestion to compare simulation results with experimental mechanical testing. However, as clarified in the revised Introduction, the aim of this study was not to validate the FEA model against experimental data, but rather to demonstrate the feasibility of a synergetic workflow that integrates micro-CT scanning, 3D printing, operator-dependent cavity preparation, and FEA. Future studies will include experimental validation and quantitative mechanical testing to build upon this foundation. We have added a dedicated "Limitations of the Study" section, highlighting the exclusion of adjacent anatomical structures (e.g., PDL and soft tissues), the potential loss of detail from polygon reduction, and the restriction of micro-CT to ex vivo use. These points clarify the scope and applicability of our current findings. The repeated use of the term "innovative" has been avoided to maintain a neutral and scientific tone. The Conclusions section was kept concise and is already presented in a structured format to clearly summarize the main contributions. We believe this format effectively communicates the findings at this stage of the research. This change can be found – page 9, from line 347
4. Response to Comments on the Quality of English Language |
|
Point 1: The English could be improved to more clearly express the research. |
|
Response 1: Thank you for the expert assessment. We have corrected the text according to the requirements of British English. |

Reviewer 2 Report
Comments and Suggestions for Authors
Integration of Micro-CT Imaging & Finite Element Simulations for Modeling Tooth-Inlay Systems
Modeling in the title but on Line 56 This predictive modelling enables dental professionals to explore stress distribution…..Please be consistent throughout the text & use double ‘ll’ for English rather than American spelling.
L 70-71 This approach allows defining an inverse task related to finding a candidate material with specific mechanical characteristics that satisfy predefined requirements for the tooth-inlay system. What is meant by ‘inverse task’? This sentIence makes little sense.
2.2.1. First micro-CT scanning
L 107 …resolution is a voxel with a size of 10 µm. It may be better to state with a cubic dimension of 10x10x10 µm.
- Results
The text has much description of methods used eg L167 Small surface features were eliminated…L175 Meshmixer was employed to correct… L178 The typodonts were printed at a layer height of 50 μm…. ALL of this paragraph is method
Please move all descriptions of the method into Methods
L 168 A surface mesh was generated and exported in STL format (Fig. 2). Fig 2 has no relatable features. The reader has no idea to which area of the tooth the image relates & the images do not look like a surface mesh. This is smoothing of the contours rather than refinement.
L 218 The geometry was discretised into 247,383 nodes and 149,953 elements… What does discretised mean?
L 222 As depicted in the leftmost image, the model was constrained on two orthogonal planes (A and C) via fixed support conditions…Is this related to Fig 5? What do A & C relate to as there is no A & C in Fig 5.
L 224 A 600 N compressive load was applied downward on the superior porous surface (point B), simulating physiological axial stress (Fig.5). Why was 600N chosen as this is not in the normal masticatory range & would be a maximum in some fully dentate individuals. Point B in fig 5 is just visible after I magnified the screen and points to a mid-lingual (?) part of the molar & not the cusp tip where loads are usually applied in function. Why was this area chosen & not the cusp tip or the
The following paragraph is the result:
L 227 The total deformation results (centre panel Fig.5) reveal a maximum displacement of 0.01788 mm. The deformation field illustrates a classic gradient profile: high deformation at the loading surface, diminishing through the porous volume toward the rigidly constrained base.
This only discusses ONE MOD inlay model. Why have the authors not presented the results for the other 3 restorations?
Discussion L257 The efficacy of region-growing algorithms in dental imaging has been corroborated by recent research. Please provide the reference.
- Conclusions This section is too long & repeats what was written in the discussion.
Overall, this text describes a technique rather than a research study. The authors have described the method in detail & provided results of the modelling on ONE typodont when 4 were prepared. The authors should describe the results of FEA loading on 4 typodonts & perhaps on different locations on the occlusal surface. This would be of interest.

Author Response
|
Comments 1: Integration of Micro-CT Imaging & Finite Element Simulations for Modeling Tooth-Inlay Systems Modeling in the title but on Line 56 This predictive modelling enables dental professionals to explore stress distribution.....Please be consistent throughout the text & use double ‘ll’ for English rather than American spelling |
|
Response 1: Thank you for pointing out the inconsistency in spelling. We have revised the manuscript to use British English spelling consistently throughout the text, including the term “modelling” with a double “l” in both the title and body of the manuscript.
|
|
Comments 2: L 70-71 This approach allows defining an inverse task related to finding a candidate material with specific mechanical characteristics that satisfy predefined requirements for the tooth- inlay system. What is meant by ‘inverse task’? This sentIence makes little sense. |
|
Response 2: Thank you for highlighting this unclear phrasing. We originally used the term "inverse task" to refer to solving an inverse problem within the context of FE analysis. Specifically, this involves formulating an optimization problem where the goal is to identify material properties that produce an FE solution (e.g., stress or strain distributions) matching predefined mechanical performance criteria or, when available, experimental data. To improve clarity and avoid confusion, we have rephrased the sentence to describe this as a design optimization strategy.
Comments 3: L 107 ...resolution is a voxel with a size of 10 μm. It may be better to state with a cubic dimension of 10x10x10 μm. Response 3: Thank you for the suggestion. We have revised the description to state the resolution as having a cubic voxel size of 10 × 10 × 10 µm and have applied this correction consistently throughout the manuscript where applicable.
This change can be found – page 3, from line 139
Comments 4: The text has much description of methods used eg L167 Small surface features were eliminated...L175 Meshmixer was employed to correct... L178 The typodonts were printed at a layer height of 50 μm.... ALL of this paragraph is method. Please move all descriptions of the method into Methods Response 4: Thank you for pointing this out. In response, we have relocated all methodological descriptions—such as mesh refinement, use of Meshmixer, and printing parameters—from the Results section to the Materials and Methods section. The Results section has been revised to focus solely on observations and outcomes.
This change can be found – page 3, from line 120
Comments 5: L 168 A surface mesh was generated and exported in STL format (Fig. 2). Fig 2 has no relatable features. The reader has no idea to which area of the tooth the image relates & the images do not look like a surface mesh. This is smoothing of the contours rather than refinement. Response 5: Thank you for your helpful observation. We have corrected the misuse of the terms “mesh” and “refinement” in the text to more accurately reflect the contour smoothing process shown in Figure 2. We have also revised the figure caption and description to clarify and improve the reader’s understanding of the image context.
This change can be found – page 6, from line 266 and page 7, line 282.
Comments 6: L 218 The geometry was discretised into 247,383 nodes and 149,953 elements... What does discretised mean? Response 6: Thank you for your question. In the context of finite element analysis, the term “discretised” refers to the numerical process of subdividing the continuous geometric domain—on which a boundary value problem is defined—into finite elements. This is a core step in the FEA, where the solution of a system of partial differential equations is approximated over a discretized domain. The resulting mesh allows the problem to be solved numerically by evaluating the equations over each element and assembling them into a global system.
Comments 7: L 222 As depicted in the leftmost image, the model was constrained on two orthogonal planes (A and C) via fixed support conditions...Is this related to Fig 5? What do A & C relate to as there is no A & C in Fig 5. Response 7: Thank you for your observation. We have revised and expanded the caption of Figure 5 to clearly indicate the reference to planes A and C, ensuring that their role in the boundary conditions is visually and contextually explained in the figure.
This change can be found – page 9, from line 341
Comments 8: L 224 A 600 N compressive load was applied downward on the superior porous surface (point B), simulating physiological axial stress (Fig.5). Why was 600N chosen as this is not in the normal masticatory range & would be a maximum in some fully dentate individuals. Point B in fig 5 is just visible after I magnified the screen and points to a mid-lingual (?) part of the molar & not the cusp tip where loads are usually applied in function. Why was this area chosen & not the cusp tip or the …… The following paragraph is the result: Response 8: Thank you for your detailed observations. The 600 N load was selected as a representative upper-bound physiological value to demonstrate the model’s structural response under maximal masticatory forces. To support this choice, we have added two relevant citations after the mention of the 600 N load in the manuscript. The mid-lingual loading point was chosen for simplicity and to ensure a consistent, controlled application across all models, rather than using cusp-tip loading, which varies anatomically and functionally. As clarified in the revised Introduction, the primary aim of this study is to introduce and validate a synergistic digital workflow for generating informative and realistic geometric and structural models suitable for numerical simulation. The FEA results are presented to demonstrate the feasibility of the workflow—not to provide a quantitative comparison between inlay designs. A full comparative analysis will be addressed in future work.
Comments 9: Discussion L257 The efficacy of region-growing algorithms in dental imaging has been corroborated by recent research. Please provide the reference. Response 9: Thank you for pointing this out. We have now added an appropriate citation to support the statement regarding the efficacy of region-growing algorithms in dental imaging.
This change can be found – page 9, line 361, Reference: 25
Comments 10: Comment on Conclusions and Scope of the Study: This section is too long & repeats what was written in the discussion. Overall, this text describes a technique rather than a research study. The authors have described the method in detail & provided results of the modelling on ONE typodont when 4 were prepared. The authors should describe the results of FEA loading on 4 typodonts & perhaps on different locations on the occlusal surface. This would be of interest. Response 10: Thank you for your thoughtful feedback. As clarified in our responses above and in the revised manuscript, the aim of this study is to present and validate a digital workflow that integrates micro-CT imaging, 3D printing, manual cavity preparation, and digital restoration design to produce realistic models suitable for FEA. While four typodonts were prepared, only one configuration was simulated and presented as a proof of concept, in line with the exploratory nature of this preliminary study. The FEA was performed to demonstrate the feasibility of the workflow, not to provide quantitative or comparative results. We agree that simulating multiple loading locations and comparing the mechanical behavior of all four inlay types would be valuable, and we have stated that this will be pursued in future work. The Conclusions section has been revised to be more concise and to avoid repeating content from the Discussion.
This change can be found – page 10, from line 420
|
|
4. Response to Comments on the Quality of English Language |
|
Point 1: The English is fine and does not require any improvement. |
|
Response 1: Thank you for the expert assessment. We have corrected the text according to the requirements of British English. |
|
|
|
|

Reviewer 3 Report
Comments and Suggestions for Authors
Dear authors,
thank you for your interesting study. There are some points that should be discussed
- Is an ethical approval needed?
- The applied tooth sample looks like a 3rd Was a periodontal problem really the reason for extraction?
- Results section is presented in a more method-like manner. Please response to this issue. What is the difference to the M&M section?
- Please discuss the choice for the occlusal pressure point.
- Describe the results for each cavity design. Are there any differences?
- In figure 5, please describe the 3 samples. What is presented?
Results about mechanical stress test by finite element analysis need to be discussed more sufficiently.
Author Response
|
Comments 1: Is an ethical approval needed? |
|
Response 1: Thank you for raising this important point. The study involved the use of one extracted human molar tooth. Informed consent was obtained from the donor for the use of the specimen in scientific research. According to our institutional and national guidelines, ethical approval is not required for in vitro studies involving anonymized, extracted teeth that are collected with consent and are not linked to any personal health data. We have clarified this point in the revised Materials and Methods section.
This change can be found – page 3, from line 122
|
|
Comments 2: The applied tooth sample looks like a 3rd Was a periodontal problem really the reason for extraction? |
|
Response 2: Thank you for pointing this out. There was a technical error in the initial submission. The tooth was extracted due to surgical indications, not periodontal disease. We have corrected this in the Materials and Methods section of the revised manuscript to accurately reflect the clinical reason for extraction.
This change can be found – page 3 , from line 122
Comment 3: Results section is presented in a more method-like manner. Please response to this issue. What is the difference to the M&M section? Response 3: Thank you for this observation. We agree that parts of the original Results section contained procedural details. In response, we have revised the manuscript to clearly separate methodology from results—moving descriptive steps to the Materials and Methods section. The Results section now focuses on key outcomes, such as the feasibility of the workflow and qualitative findings from the finite element analysis
This change can be found – page 3, from line 120 and page 6, from line 251.
Comment 4: Please discuss the choice for the occlusal pressure point. Response 4: Thank you for this comment. The pressure point was deliberately placed on the mid-lingual surface of the occlusal plane rather than a cusp tip to apply a broad-area compressive load and avoid highly localized contact artifacts in this preliminary study. This setup allowed us to evaluate overall stress distribution and deformation patterns within the restored system. We acknowledge that cusp-tip loading reflects more realistic occlusal function and plan to include such localized and dynamic loading scenarios in future work for comparative analysis.
This change can be found – page 8, from line 322
Comments 5: Describe the results for each cavity design. Are there any differences? Response 5: Thank you for your observation. In this preliminary study, the primary aim was to introduce and validate the digital workflow integrating micro-CT imaging, 3D printing, manual preparation, and FEA. Although four cavity designs were created and prepared, only one representative model was subjected to finite element simulation to demonstrate the feasibility and realism of the workflow. As such, comparative mechanical results between the cavity types are not included in the current manuscript. We fully agree that such an analysis would provide valuable insights, and we plan to include full quantitative comparisons between the different designs in future investigations.
This change can be found – page 6, from line 251
Comments 6: In figure 5, please describe the 3 samples. What is presented? Response 6: Thank you for pointing this out. Figure 5 has been updated with a more detailed caption to clarify the content. The figure presents:
This change can be found – page 9, from line 341
Comments 7: Results about mechanical stress test by finite element analysis need to be discussed more sufficiently. Response 7: Thank you for pointing this out. You can find this in section Discussion.
This change can be found – page 9, from line 347.
|
|
4. Response to Comments on the Quality of English Language |
|
Point 1: The English is fine and does not require any improvement |
|
Response 1: Thank you for the expert assessment. We have corrected the text according to the requirements of British English. |
|
|
|
|

Round 2
Reviewer 2 Report
Comments and Suggestions for Authors
The script is much improved.